

# High-resolution long-term average groundwater recharge in Africa estimated using random forest regression and residual interpolation

Anna Pazola[1,2], Richard G. Taylor[1], Mohammad Shamsudduha[3], Jon French[1], Alan M. MacDonald[4], Tamiru Abiye[5], and Ibrahim Baba Goni[6]

[1]Department of Geography, University College London, Gower St, London WC1E 6BT, United Kingdom
[2]United Nations Environment Programme World Conservation Monitoring Centre, 219 Huntingdon Road, Cambridge CB3 0DL, United Kingdom
[3]Institute for Risk & Disaster Reduction, University College London, Gower St, London WC1E 6BT, United Kingdom
[4]British Geological Survey, Lyell Centre, Research Avenue South, Edinburgh EH14 4AP, United Kingdom
[5]School of Geosciences, University of the Witwatersrand, 1 Jan Smuts Ave, Braamfontein, Johannesburg 2000, South Africa
[6]Department of Geology, University of Maiduguri, 1069 Bama - Maiduguri Rd, 600104, Maiduguri, Nigeria

**Correspondence:** Anna Pazola (anna.pazola.20@ucl.ac.uk)

**Abstract.** Groundwater recharge is a key hydrogeological variable that informs the renewability of groundwater resources. Long-term average (LTA) groundwater recharge provides a measure of replenishment under the prevailing climatic and land-use conditions and is therefore of considerable interest in assessing the sustainability of groundwater withdrawals globally. This study builds on the modelling results of MacDonald et al. (2021) who produced the first LTA groundwater recharge map

across Africa using a linear mixed model (LMM) rooted in 134 ground-based studies. Here, continent-wide predictions of groundwater recharge were generated using Random Forest (RF) regression employing five variables (precipitation, potential evapotranspiration, soil moisture, NDVI and aridity index) at a higher spatial resolution ($0.1°$ resolution) to explore whether an improved model might be achieved through machine learning. Through the development of a series of RF models, we confirm that a RF model is able to generate maps of higher spatial variability than LMM; the performance of final RF models in terms

of the goodness of fit ($R^2 = 0.83$, $0.88$ with residual kriging) is comparable to the LMM ($R^2 = 0.86$). The higher spatial scale of the predictor data ($0.1°$) in RF models better preserves small-scale variability from predictor data, than the values provided via interpolated LMM; these may provide useful in testing global-to-local scale models. The RF model remains, nevertheless, constrained by its representation of focused recharge and by the limited range of recharge studies in tropical Africa, especially in the areas of high precipitation. This confers substantial uncertainty in model estimates.

## 1   Introduction

Groundwater is the largest store of unfrozen freshwater on Earth and enables vital, climate-resilient access to water for drinking, agriculture and industry (Müller Schmied et al. (2021)). Across Africa, most rural and many urban communities are strongly dependent on groundwater, especially in the arid and semi-arid regions where it is often the only perennial source of water (UNEP (2010); Gaye and Tindimugaya (2019)). Groundwater resources are unevenly distributed across the African

continent and are characterised primarily by two aquifer systems: low-recharge/high-storage regional sedimentary aquifers and



high-recharge/low-storage weathered crystalline rock aquifers (MacDonald et al. (2021)). Freshwater demand is projected to increase substantially in pursuit of the United Nations Sustainable Development Goal 2 (zero hunger) and 6 (water and sanitation for all) among others. Only ~30% of the population of Africa has access to safe drinking water (WHO and UNICEF (2021)) and less than 5% of the arable land is irrigated (Siebert et al. (2010); Villholth (2013)). Calls to increase groundwater abstraction across Africa (e.g. Calow et al. (2010); Altchenko and Villholth (2015); Gaye and Tindimugaya (2019); Olago (2019); Cobbing and Hiller (2019)) are growing to support economic development according to the United Nations Agenda 2030 Sustainable Development Goals (Guppy et al. (2018)).

Recharge is the downward flow of water that reaches the saturated (phreatic) zone and contribute to aquifer storage (De Vries and Simmers (2002)). Groundwater recharge is often assumed to be diffuse, derived from the direct or near-direct infiltration of rainfall at the soil surface through the landscape. Recent research has highlighted, however, the importance of focused recharge in African drylands (Cuthbert et al. (2019); Seddon et al. (2021); Goni et al. (2021)), which takes place via leakage from ephemeral streams and ponds. The definition of what constitutes renewable groundwater resources varies (Gleeson et al. (2020)) but the long-term average (LTA) groundwater recharge provides a measure of aquifer replenishment under the prevailing climatic and land-use conditions and is therefore of considerable interest in assessing the sustainability of groundwater withdrawals not only in Africa but globally.

A range of climatic, hydrological and hydrogeological variables influence groundwater recharge fluxes (e.g. Van Wyk et al. (2011); Mohan et al. (2018); Moeck et al. (2020)). Precipitation and potential evapotranspiration and their seasonal variability have the biggest influence, as they directly affect the initial amount of water available for recharge. Some studies estimate that precipitation alone can explain 80% of the variation in groundwater recharge (Keese et al. (2005)). Recently, Berghuijs et al. (2022) showed that much of the variations in groundwater recharge can be explained by a sigmoidal function of climate aridity and precipitation. Vegetation influences important processes such as infiltration rates, deep drainage and effective rainfall. Consequently, changes in land cover can lead to substantial variations in groundwater recharge (Scanlon et al. (2006); Favreau et al. (2009)). Also, the root-zone saturation impacts the distribution of soil hydraulic conductivity and affects both the percolation of water to the groundwater table and water uptake by plant roots (O'Geen (2013)). Due to the complexity of the processes influencing recharge, other parameters related to the aforementioned factors are identified by regional and global scale studies as important recharge factors, including seasonality in temperature, depth to the water table, elevation, slope, and soil texture (Nolan et al. (2007); Mohan et al. (2018); Moeck et al. (2020)).

Large-scale estimates of groundwater recharge typically involve the use of mechanistic models (Doell and Fiedler (2007); Wada et al. (2010); Koirala et al. (2012)). The accuracy of these models suffers from knowledge gaps in the relationships among recharge and topographical, lithological and land-cover factors (Mohan et al. (2018)) and inadequate representation of focused recharge, which is the main source of aquifer replenishment in semi-arid and arid areas (Taylor et al. (2013); Cuthbert et al. (2019)). Recent developments of the WaterGAP Global Hydrological Model incorporate groundwater recharge below surface water bodies and improved rules for the conditions under which water remains in the soil instead of becoming surface runoff in semi-arid and arid regions (Müller Schmied et al. (2021)), with a planned integration of gradient-based groundwater model to further incorporate focused recharge (Reinecke et al. (2019)). Still, global- and continental-scale models are





tested at eco-region, climatic region or large river basin scales and are untested by recharge observations, leading to consider-able inaccuracies in groundwater recharge estimates, which are currently addressed by tuning parameters in WaterGAP v2.2d (Müller Schmied et al. (2021)).

Data-driven empirical models can be developed as an alternative to process-driven physical models, as they bypass the cur-rent knowledge gaps concerning the processes governing long-term groundwater storage and recharge. Such an approach was recently followed by MacDonald et al. (2021) who employed a linear mixed model (LMM) to map groundwater recharge across Africa for the first time using a curated database of long-term ground-based recharge observations. Their results demonstrate that long-term mean annual precipitation is by far the strongest predictor of long-term groundwater recharge. In combination with the outcome of a previous study on groundwater storage (MacDonald et al. (2012)), the perception of water scarcity across Africa can be reevaluated, as most countries with little groundwater storage experience high groundwater recharge, whereas most arid areas in Africa with negligible precipitation are located above regional sedimentary aquifers. As a result of these studies, the areas of renewable and non-renewable groundwater resources can be identified, which can inform sustainable water use.

## 1.1 Data-driven methods for groundwater modelling

The LMM technique employed by MacDonald et al. (2021) is a well-established statistical approach for regression in life sciences and beyond (e.g. Harrison et al. (2018)) that is able to handle multicollinearity of covariates. However, it requires careful fitting and makes several assumptions about the distribution of errors. There has been a growing interest in the potential of machine learning (ML) methods as an alternative to statistical models in the field of groundwater modelling. ML methods such as artificial neural networks, support vector machines and decision trees have been applied to predict groundwater levels (Bowes et al. (2019)), map groundwater contamination (Podgorski and Berg (2020)) and identify groundwater potential zones (Al-Fugara et al. (2020)). A recent study by Huang et al. (2019) employed multi-layer perception network and deep learning to predict a time series of annual average groundwater recharge on a regional scale in Australia. Although such models operate in a black-box manner and have no explanatory power with regards to the underlying physical processes, they can often deliver accurate predictions. These are, however, limited by the choice and quality of forcing data, as well as by the availability of measurements necessary for model training and testing. Additionally, different ML methods pose different challenges in their application. For example, neural networks require careful choice of hyperparameters, logistic regression lacks sensitivity toward outliers and support vector machines can only be applied to independent and identically distributed input data. These limitations can be avoided using modern ML methods that include ensemble learning algorithms that average a series of predictions to create a more robust final model, such as random forest (Breiman (2001)).

The random forest (RF) technique is based on a series of classification or regression decision trees whose individual pre-dictions are averaged to create a unique model. Of note is that it can handle complex interactions between variables, mul-ticollinearity and non-linearity of predictors. Due to its non-parametric nature, it does not require extensive hyperparameter tuning. In the field of groundwater modelling. The RF technique has been successfully applied to map arsenic contamination globally (Podgorski and Berg (2020)) and nitrate concentrations in groundwater across Africa (Ouedraogo et al. (2018)).



## 1.2 Aims

The aims of this study are: (1) to test the results of the continental-scale 0.5° spatial resolution LMM of groundwater recharge (MacDonald et al. (2021)) against a data-driven random forest (RF) regression model; (2) to develop a higher resolution (0.1°) continental-scale groundwater recharge RF model; and (3) to compare recharge maps obtained using both approaches at 0.5° and 0.1° resolutions.

The first aim is achieved by fulfilling the following tasks:

- Revisiting the study of MacDonald et al. (2021) and the acquisition of datasets of explanatory factors at an appropriate resolution;

- Training a RF model and mapping of LTA groundwater recharge at a spatial resolution of 0.5° for the time period 1981–2010; and

- Comparing model performance and spatial differences in predicted groundwater recharge patterns across the African continent.

The second aim is achieved through:

- Collation of datasets for explanatory factors at a higher spatial resolution of 0.1°; and

- Training of a RF model and producing a map of LTA groundwater recharge at a spatial resolution of 0.1°.

The final aim involves:

- Development of a linear mixed model at a spatial resolution of 0.1°; and

- Comparison of predicted LTA groundwater recharge between different models.

Section 2 summarises the study area and the spatial characteristics of its groundwater resources, and outlines the data sources and the model development process. Section 3 presents the results of the modelling experiments. Section 4 discusses these results in the wider context and critically evaluates the developed model. This study is accompanied by a Supporting Material that provides extensive information on the predictors used and additional analyses that extend the investigation presented in this paper.

## 2 Materials and methods

### 2.1 Study area

The occurrence of groundwater resources and their accessibility in continental Africa is conditioned by geology, geomorphology as well as historic and current climatic conditions. There are four main hydrogeological environments across the continent: crystalline basement, consolidated sedimentary rocks, unconsolidated sediments and volcanic rocks, which occupy 34%, 37%,





25% and 4% of the land area respectively (MacDonald and Calow (2009); Fig. S2, Supplementary Material). However, there are significant variations in the characteristics of groundwater resources between and within these environments. Aquifers
within crystalline basement rocks found primarily across equatorial Africa are shallow with generally low yields. In contrast, regional sandstone aquifers in the Sahara contain enormous groundwater volumes that originate from wet climatic periods in the late Pleistocene and early Holocene (Abouelmagd et al. (2012)).

Rainfall is highly variable across the continent due to the interactions of continental tropical, maritime equatorial and maritime tropical air masses in the intertropical convergence zone (Van Wyk et al. (2011)). These provide a basis for the division of
the continent into 8 climatic regions, most of which experience high interannual rainfall seasonality. Mean annual precipitation varies from negligible across the Sahara to very high rates in equatorial regions, notably ~10 000 mm/yr in the Gulf of Guinea.

## 2.2 The groundwater recharge dataset for Africa

The dataset of ground-based groundwater recharge measurements used for model fitting and testing was compiled by MacDonald et al. (2021). It aggregates recharge estimates from various published and grey studies across the continent, including direct
and indirect field measurements obtained using common methods: chloride mass balance, environmental and isotropic tracers, groundwater-level fluctuation and soil moisture balance methods, as well as modelled recharge values reconciled to field data. The existing online databases were critically reviewed and assigned confidence ratings ranging from 1 (high confidence) to 5 (low confidence). Out of 316 identified studies, 134 sample points were selected for the final dataset, with the majority of entries classified as "medium confidence" (ranks 2-4) and only 4 points that obtained the lowest confidence score. This com-
pilation thus constitutes the most robust dataset yet of multi-decadal estimates of distributed natural groundwater recharge for the period 1970-2019. It primarily comprises estimates of diffuse recharge, but may, in places, include recharge from focused pathways; studies of focused recharge from surface water bodies, ephemeral overland flow, urban leakage and irrigation returns were explicitly excluded.

Given that this investigation examines modern recharge and the renewability of groundwater resources using a data-driven
approach that assumes a causal link to the set of predictors, a few data points representing no rainfall-fed recharge, primarily in deep fossil North-Eastern Saharan basins, were excluded from most of this analysis. As a result, 127 estimates from the original dataset were used in all but one experiment that investigated the impact of the inclusion of these zero-recharge samples on the recharge prediction when compared to the initial map. The spatial distribution of groundwater recharge observational points is shown in Figure 1. There is a visible inequality in the representation of different climatic zones. Southern Africa has
good data coverage whereas central Africa, including the Congo basin, is data-sparse. Notably, only a few measurements in the drought-prone Horn of Africa are available, where the aridity of the climate, high water stress and the recent decrease in rainfall fuel projected high dependence on groundwater (Funk et al. (2015); Thomas et al. (2019)).

## 2.3 The dataset of explanatory factors

Predictors related to climate, land use, soil type and hydrogeology, in particular those used by MacDonald et al. (2021), were
considered to have the highest explanatory power in estimating groundwater recharge. According to the earlier study, precipi-



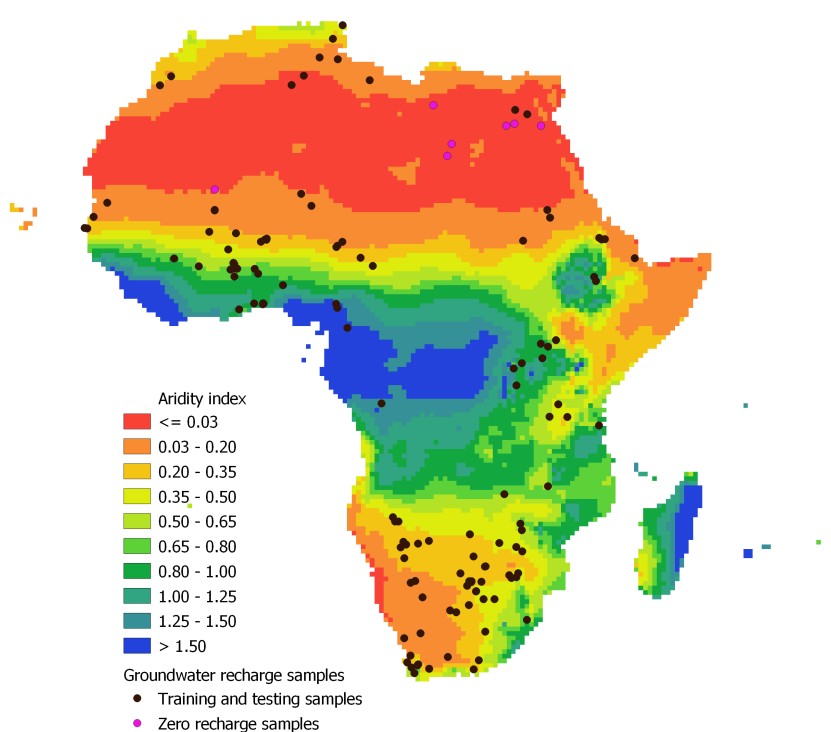

**Figure 1.** Data samples used for model training and testing, alongside zero-recharge points omitted in the initial variable importance analysis, compiled by MacDonald et al. (2021) in relation to the aridity of the region. Aridity index data obtained from CGIAR-CSI (https://cgiarcsi. community/data/global-aridity-and-pet-database/, accessed on 28.07.2021)

tation dominates all other signals. Consequently, the same dataset on precipitation for the time period 1981-2010, originating from Climate Research Unit gridded Time Series (CRU TS) (Harris et al. (2020)) was used to replicate its results. CRU TS is created by interpolating monthly climate anomalies obtained from numerous international weather stations, giving a global gridded dataset at a spatial resolution of $0.5°$. Gridded data on remaining explanatory factors (MacDonald et al. (2021))

were obtained from the same sources as in the original analysis. Potential evapotranspiration and aridity index were provided alongside precipitation data by CRU TS dataset. NDVI data are provided by NASA's Moderate Resolution Imaging Spectroradiometer (MODIS) satellite product at a spatial resolution of $0.05°$. Aquifer domain data were obtained from an earlier study by the British Geological Survey (MacDonald et al. (2012)); soil group information derives from the Soil Atlas of Africa developed by a joint research effort of the European Union and FAO (Jones et al. (2013)). Land cover data were extracted from

the ESA's GlobCover satellite product at a resolution of 300m. Number of wet days, used as a covariate in predictor selection analysis by MacDonald et al. (2021), was omitted in this analysis due to its low explanatory power in the initial analysis at $0.5°$ and difficulties in obtaining a dataset at a higher resolution. Additionally, based on the latest literature findings, a range





of other factors was identified as potentially insightful for this investigation (Mohan et al. (2018); Moeck et al. (2020)). Of these, two variables: elevation and soil moisture, were incorporated into the models. The corresponding datasets were obtained

from NASA's Shuttle Radar Topography Mission (SRTM) digital elevation version 4 and Famine Early Warning Systems Network Land Data Assimilation System (FLDAS) Noah Land Surface Model L4 products. Using meteorological variables from MERRA-2 analysis as forcing data, the FLDAS model, Noah 3.6.1, produced global estimates of land surface variables at a spatial resolution of $0.1°$.

In order to create a groundwater recharge map at a spatial resolution of $0.1°$, additional datasets of higher resolution from the

Consultative Group on International Agricultural Research - Consortium for Spatial Information (CGIAR-CSI) are employed for potential evapotranspiration and aridity. They are both modelled at a spatial resolution of $0.01°$, using long-term monthly averaged climate data from WorldClim. Also, another precipitation dataset of a higher resolution (Climate Hazards Group InfraRed Precipitation with Station data (CHIRPS) version 2 at a spatial resolution of $0.05°$) is used for developing both RF and LM models. CHIRPS data are produced by combining models of terrain-induced precipitation enhancement with

interpolated ground-based station data and gridded satellite-based precipitation estimates from NASA and NOAA (Funk et al. (2015)). A summary of the data sources for each variable, alongside the maps of spatial distributions of all explanatory factors is given in Table S1 in Supplementary Material.

### 2.4 Random forest model

Random forest (RF) is an ensemble machine learning method for classification and regression based on randomised decision

trees (Breiman (2001)). The fundamental concept behind the algorithm is that an average of the prediction probabilities of a large number of models outperforms any of the individual models in terms of accuracy. Consequently, RF assembling an average model out of a large number of decision trees, is superior to just one model (one decision tree), which is very sensitive to outliers, unstable and tends to overfit. RF inherits most of the advantages of decision trees: the ability to handle both numerical and categorical input data, the lack of need for data preparation such as input data normalisation, and robustness

against multicollinearity of features. The theory and technical description of a decision tree algorithm, and subsequently the random forest regression, is introduced by Breiman et al. (1984) and Breiman (2001).

When training a RF model, every contributing decision tree is built based on a subset of training data drawn with replacement (bagging), so that the same sample can occur multiple times or might be omitted in a single tree creation process. About two-thirds of these samples are used for training, while the remaining "out-of-bag" (oob) samples are used for internal cross-

validation, resulting in an oob performance score for the entire random forest model (Breiman (2001)). Additionally, the nodes in an individual tree are split using the best split predictive variable from a selected subset of features that changes randomly across different trees. The actual tree training data, being a subset of the original model training set used for creating a RF, vary in the number of employed predictors and in the number and composition of training samples across different decision trees. As a single tree tends to overfit its training data set, an average over a high number of decision trees greatly reduces

the prediction variance and delivers a model that is relatively robust to outliers which could distort the performance of other algorithms such as neural networks.




## 2.5 Linear mixed model

The statistical model tested against the random forest model is a linear mixed model (LMM), a well-established method that is suitable for handling spatially dependent data. MacDonald et al. (2021) used this approach to generate a continental LTA groundwater recharge map at a spatial resolution of 0.5°. Here, their procedure is replicated to obtain a map at a higher resolution of 0.1° and to allow for the result-based comparison of RF and LM models at both resolutions, LMM is an extension of a simple linear model that allows for both fixed and random effect terms. While fixed effects comprise all predictors, which have a fixed relationship with the response variable across all observations, random effects account for the fact that fixed effects are expected to be spatially dependent, as observations in one area are likely to be more similar than those further apart. The map generation procedure closely follows the steps from MacDonald et al. (2021), i.e. the LMM is used to compute the empirical best linear unbiased prediction (E-BLUP) of LTA recharge at unsampled sites on a prediction grid. As the locations of LTA recharge observations exhibit spatial dependence, E-BLUP combines the predicted value from the fixed effects and interpolated random effects to account for the variability among observation clusters and to minimise the expected prediction error variance. Further details on the developed model are described in Section 3 of Supplementary Material and MacDonald et al. (2021), whereas the theoretical description of LMM can be found in Lark et al. (2006).

## 2.6 Residual kriging

Residual kriging is applied as a part of the procedure to compute E-BLUP of LTA recharge at a spatial resolution of 0.1° to minimise the expected squared prediction error. As the input data values for the LMM at 0.1° come from related predictor datasets as used for the LMM by MacDonald et al. (2021) and the residuals most likely exhibit similar patterns, the same assumptions are made regarding the choice of initial variogram parameters, covariance function and model fitting method. The final interpolation setup, including optimised model parameters, is described in Section 3 of Supplementary Material. Residual kriging is also applied on top of the prediction results from the RF model. As algorithmic modelling (unlike parametric modelling) does not make any assumptions about the underlying process from which the observations originate, the spatial dependence of observations is not taken into consideration at any point by the base RF model. However, we explicitly account for variability in LTA recharge observations around the fitted values by investigating the residuals and performing kriging-type interpolation so that the correction of RF results is carried out under the assumption that the LTA recharge observations are likely to be similar for all unsampled sites in the proximity to the sampled sites.

## 2.7 Model development

The computer code developed in this analysis is written in Python 3.9 and R 4.2.0 and is available on Github at https://github.com/pazolka/rf-groundwater-recharge-africa. The main code is partitioned into multiple Jupyter Notebooks the names of which correspond to the names of the major steps of the analysis. The processed input raster files derived from data freely accessible online, are available through an open repository linked to this paper.





The random forest implementation used in this study is *RandomForestRegressor* from Python library *scikit-learn* version 0.24.2. To generate a LMM based recharge map at $0.1°$, the steps described by MacDonald et al. (2021) were followed using R library *spaMM* version 3.11.14. Residual interpolation applied in addition to RF and LMM results was performed using R library *gstat* because of more advanced variogram fitting options.

For interactive inspection of LTA recharge maps generated using developed models, the results were visualised using a Python package *geemap* (Wu et al. (2019); Wu (2020)).

### 2.7.1 Input data processing

All data were obtained in the form of GeoTIFF raster files (WGS84 projection), using both data provider' websites and the Google Earth Engine platform. Time series for precipitation and soil moisture were averaged for the period 1981-2010 to obtain gridded long-term mean values commensurate with the temporal resolution of recharge observational data. Where necessary, input grids were rescaled to an appropriate resolution of $0.5°$ or $0.1°$. Continuous point data were upscaled using bilinear interpolation, whereas mode resampling method was applied to categorical data.

To ensure the consistency of the training data, the predictor values were sampled at each groundwater recharge observational point, resulting in two input datasets, one for each spatial resolution of interest.

### 2.7.2 Training and testing datasets for random forest model

RF is a supervised ML algorithm. It uses labelled training data to construct a function inferring the desired output value from an input object in a process called model training. Apart from the training dataset, a testing dataset is typically employed to assess the ability of the algorithms to generalise from the training values to unseen data. As the RF model employs a form of internal cross-validation and builds each decision tree using only a subset of the training data, as outlined in Section 2.4, all recharge samples were used to build the final models used for predicting recharge values on the continental scale. In that step, model validation using a separate testing set was omitted and the out-of-bag score was used to check generalisation ability of the final models. However, explicit partition of data into training and testing data consisting of 88 and 39 recharge samples (70% and 30% respectively, following Nguyen et al. (2021)) was performed for different subtasks in this study that required external validation, such as hyperparameter optimisation and performance assessment.

### 2.7.3 Performance metrics

The goodness of fit of models is primarily evaluated by calculating the coefficient of determination $R^2$ of the modelled and observed values, expressed as:

$$R^2 = 1 - \frac{\sum_{i=1}^{n}(y_i - \hat{y}_i)^2}{\sum_{i=1}^{n}(y_i - \bar{y})^2} \tag{1}$$

where $y_i$ is the observed $i$-th value, $\hat{y}_i$ is the predicted $i$-th value, $\bar{y}$ is the mean value. Also, out-of-bag (oob) performance score, a random forest-specific metric of internal model validation, is investigated after each model training. $R^2$ is a measure of goodness of fit in capturing the variance and is expressed in relative values, therefore root mean squared error (RMSE) is



employed to quantify the absolute fit of the model. Apart from calculating the quantitative performance metrics, a visual analysis of a plot of observed versus modelled values, and plots of observed and modelled values versus residuals were undertaken to inspect the presence of possible problems in the underlying data.

### 2.7.4 Data transformation

The LMM assumes the normality of residuals, so that log-transformation of LMM input data was required. The RF algorithm is non-parametric and makes no assumptions about the underlying statistical nature of the data. Predictive variables may be
numerical or categorical, follow any distribution and have different scales, requiring no extensive transformation. Although non-parametric models are rarely affected by skewness in the dependent variable, transforming the response variable can lead to predictive improvement in some cases (Boehmke and Greenwell (2019)). A preliminary check suggested that log-transformation does not significantly improve the predictive ability of the random forest algorithm. However, to increase the prediction performance of low recharge values and to make the treatment of dependent variable consistent across the models,
log-transformation was applied to RF input data as well. The output data were back-transformed to the original scale. A similar transformation procedure was applied in other random forest applications (e.g. Wheeler et al. (2015); Ouedraogo et al. (2018)).

### 2.7.5 Hyperparameter tuning

The RF model has two important user-defined hyperparameters: the number of decision trees and the number of randomly selected predictors used to split the nodes. Optimisation of these parameters can significantly reduce the generalisation error
(Breiman (1996); Peters et al. (2007)). Concerning the number of trees, this value can be as large as possible since RF does not overfit and is computationally efficient and parallelisable (Breiman (2001); Probst and Boulesteix (2017)); this study utilised 2000 decision trees. The recommended number of randomly selected predictors for each node split in a regression tree is the number of all predictors divided by three (Breiman (2001); Hastie et al. (2009)). Other minor hyperparameters were tuned using random search technique with 3 fold cross-validation across 100 different combinations. Their values are summarised
in Table 1. The hyperparameters were optimised again when adding zero-recharge samples to the recharge dataset and when using input data at a spatial resolution of $0.1°$ in further analysis.

### 2.7.6 Random forest models for LTA groundwater recharge in Africa

Complementary to the development of a RF model for continental LTA groundwater recharge, a simple predictor importance analysis using the RF's built-in feature importance, was performed (see Section 5a of Supplementary Material). As the result,
the predictor list for the final RF models used to generate maps was reduced to contain five variables (precipitation, soil moisture, NDVI, potential evapotranspiration and aridity index), as land cover, aquifer group, soil group and elevation were found to have a negligible effect on the explanatory power of the model.

The creation of a continental recharge map at a spatial resolution of $0.5°$ incorporated the selected variable set and the optimised hyperparameters. A single RF model was trained using all available samples. Finally, recharge values were predicted



for the whole domain, and model performances in terms of $R^2$ and $RMSE$ values of both the RF model and the LMM by MacDonald et al. (2021) were compared. Additionally, the absolute and relative spatial differences in recharge estimates between the models were obtained and investigated. A similar procedure was applied to obtain continental recharge values at a spatial resolution of $0.1°$ using higher resolution predictor data, with an additional step needed to create an LMM at $0.1°$ by replicating the procedure by MacDonald et al. (2021).

Another pair of LTA recharge maps - at $0.5°$ and $0.1°$ - was created by extending the results of base RF models and interpolating the residuals from the RF predicted value and the observed LTA recharge. Sections 3.2.2 and 3.3.3 compare the continental maps generated by base and kriged RF models, and the LMM at the respective resolution.

Although samples within aquifers where no modern recharge was detected were explicitly excluded from the variable sensitivity analysis and from the principal recharge modelling due to the assumed lack of causality between the predictors and the recharge, another recharge map was created to assess the influence of the inclusion of zero-recharge samples. As the previously found optimal hyperparameters might have become suboptimal after the inclusion of 7 extreme values, hyperparameter tuning was performed again. A new RF model was trained using all 134 recharge samples and applied to the feature data of the whole domain to predict the recharge. Finally, the absolute and relative spatial differences between the recharge map obtained using the principal model and the model including zero-recharge samples were investigated. These results are presented in Section 3.2.1.

Additionally, a supporting precipitation sensitivity analysis was carried out to investigate the influence of different input precipitation datasets on the continental LTA recharge map at the spatial resolution of $0.5°$. The results confirmed that the choice of the precipitation data source has a significant impact on predicted values, as the precipitation signal is dominant over other predictors and the RF model directly reflects the spatial distribution of the precipitation data and accentuates differences between the individual datasets (see Section 5b of Supplementary Material).

## 3  Results

To check the initial performance of the RF model before hyperparameter optimisation, a series of 100 models was built. The $R^2$ values of training and testing sets ranged between $0.94$ and $0.96$ and $0.28$ and $0.80$ respectively. The oob score oscillated between $0.57$ and $0.73$. It was possible to trade some accuracy on the training set for more accuracy on the testing set through tuning minor hyperparameters limiting trees capacity. Table 1 lists the optimal hyperparameters found through random search with cross-validation.

After applying the aforementioned hyperparameters to another series of 100 RF models, $R^2$ values of the training and testing sets ranged between $0.8$ and $0.84$ and $0.52$ and $0.86$ respectively. The oob score oscillated between $0.59$ and $0.73$. The optimised hyperparameters were used for generating recharge maps at spatial resolutions of $0.5°$ and $0.1°$.





## 3.1  Evaluation of model performance on training and testing data

In addition to performance assessment based on the quantitative metrics (summarised in Table 2), a visual analysis of model residuals was performed based on a single model selected randomly out of the series of models E2. The model $R^2$ values were 0.85 and 0.59 for the training and testing datasets respectively, and the oob value was 0.69. The residual and predicted vs. observed plots across training and testing sets are included in Section 5c of Supplementary Material. High recharge values are
mostly underestimated in both sets, as there are only a few measurements in the areas of high precipitation and high recharge. Also, some low recharge values are overestimated on a relative scale.

Several groundwater recharge sample points from the training and the testing set exhibit significant discrepancies between their observed and predicted values. Notably, this includes all samples obtained from Burkina Faso, situated in the semi-arid and tropical wet-and-dry climate zones. The model underestimates these samples (136 obs/38 pred, 221 obs/64 pred, 266

**Table 1.** Optimal random forest hyperparameters found through random search with cross-validation for different random forest model variants used in this study.

| Hyperparameter name | Description | RF 0.5° | RF 0.5° (+ zero recharge) | RF 0.1° |
|---|---|---|---|---|
| n_estimators | number of trees in the forest | 2000 | 2000 | 2000 |
| min_samples_split | min number of data points placed in a node before the node is split | 10 | 10 | 15 |
| min_samples_leaf | min number of data points allowed in a leaf node | 1 | 1 | 1 |
| max_features | max number of features considered for splitting a node | 0.33 | 0.33 | auto |
| max_depth | max number of levels in each tree | 80 | 80 | 60 |
| bootstrap | method for sampling data points (with or without replacement) | True | True | True |

**Table 2.** Performance results of multiple random forest ensemble runs, each consisting of 100 random forest models, for both training and testing datasets. $R^2$ values, calculated for both log-transformed and back-transformed results, are expressed as an ensemble range, with the average value across all runs indicated within parentheses. RMSE values are calculated in the original scale for the best single model run in the series. Y refers to the target variable: groundwater recharge.

| Model ensemble | $R^2$ train (log scale) | $R^2$ test (log scale) | $R^2$ train (orig. scale) | $R^2$ test (orig. scale) | Best train RMSE (orig. scale) | Best test RMSE (orig. scale) | oob score | Description |
|---|---|---|---|---|---|---|---|---|
| E1 | 0.93-0.95 (0.94) | 0.24-0.83 (0.60) | 0.71-0.87 (0.80) | -2.98-0.75 (0.11) | 63.35 | 60.65 | 0.54-0.72 (0.63) | log(Y) |
| E2 | 0.81-0.86 (0.84) | 0.34-0.87 (0.63) | 0.43-0.61 (0.53) | -0.41-0.81 (0.36) | 85.49 | 26.64 | 0.60-0.72 (0.67) | log(Y), tuned hyperparameters |
| E3 | 0.79-0.88 (0.83) | 0.19-0.77 (0.64) | 0.43-0.72 (0.53) | -0.28-0.59 (0.34) | 85.95 | 39.77 | 0.58-0.76 (0.66) | log(Y) zero-samples included, tuned hyperparameters |



obs/126 pred), which may indicate the influence and importance of focused recharge in this area. A similar conclusion applies
to two observational points in the semi-arid and arid regions in Ethiopia (185 obs/19 pred and 167 obs/24 pred). In general, the
model understates extremely high recharge values in the tropical regions of DR Congo, Cameroon and Benin. The recharge
observations amount to 420, 941 and 491 mm/yr whereas modelled values were 123, 125 and 167 mm/yr, which results in
a few large residuals driving down the overall model performance. Interestingly, the predicted recharge for an observational
point in Uganda, situated in the highlands, overestimates the recharge (17 obs/67 pred), possibly due to the scarcity of data
obtained from similar environments, and the absence of elevation-related variables in the set of explanatory factors.

**Table 3.** Performance of random forest models used to predict groundwater recharge on the continental scale at a spatial resolution of $0.5°$. Obtained $R^2$ values refer to the training set consisting of the entire available recharge sample data in the log scale, including or excluding the zero-recharge points. Metrics of the linear mixed models are included for comparison.

|  | Model | $R^2$ obs. vs pred. | $R^2$ RF vs LMM | Out-of-bag score | Description |
|---|---|---|---|---|---|
| $0.5°$ | rf | 0.83 | 0.94 | 0.68 | zero-recharge points excluded |
|  | rf_zeros | 0.85 | 0.96 | 0.76 | zero-recharge points included |
|  | rf_rk | 0.88 | 0.94 | - | RF + residual kriging |
|  | lmm | 0.86 | - | - | MacDonald et al. (2021) |
| $0.1°$ | rf | 0.80 | 0.91 | 0.65 | zero-recharge points excluded |
|  | rf_rk | 0.87 | 0.95 | - | RF + residual kriging |
|  | lmm | 0.92 | - | - | this study |

## 3.2 Modelling groundwater recharge across Africa at 0.5° spatial resolution

Performance of the base RF model used for the groundwater recharge modelling on the continental scale is presented in Table
3. As the model was trained using all non-zero groundwater recharge sample points, its generalisation ability was assessed
based exclusively on the oob score. Prediction performance on the training set, in terms of the $R^2$ value in the log scale, was
compared with the performance of the LMM by MacDonald et al. (2021). Both models fit the observed recharge with similar
results. The RF model was able to reproduce the results and, in some cases, provide marginally better predictions, e.g. for
the recharge values $200 < Y < 500$, as illustrated in Figure S10 in Supplementary Material. The satisfactory out-of-bag value
indicated that the model did not overfit the training data. However, both models did not generalise well for high recharge values.
The combined model consisting of a RF model and residual kriging shows an improved fit in terms of $R^2$ value (0.88 vs
0.83), but the effect is very localised, as the spatial dependence of residuals based on the fitted variogram is restricted to around
180-200 km.





### 3.2.1 Inclusion of zero-recharge samples

The inclusion of zero-recharge samples increased the overall fit of the RF model to observations ($R^2$ = 0.83 vs 0.89). However,
this effect was caused predominantly by the inclusion of samples whose residuals were relatively small, driving the overall
error down and thus increasing the score. Therefore, the RMSE values of non-zero recharge samples in both RF models were
compared. A reduction from 79.5 mm/year to 76.5 mm/year suggests that the inclusion of zero-recharge samples did not
negatively influence the recharge predictions of the remaining samples. The increased out-of-bag score confirmed that the
model generalisation ability did not get compromised. The overall agreement of the LM and RF models improved as well.

As illustrated in Figures 2d and 3, the inclusion of zero-recharge samples mostly impacted recharge predictions in hyper-arid
regions of North Africa, although in absolute terms this was equivalent to a decrease from less than 2 mm/yr to nearly zero
recharge. It also contributed to modelled values being up to 10 to 20% higher in the tropics, in particular in the Congo Basin.
The modelled values across other parts of Africa remained mostly unchanged. Curiously, there is a slight rise in the recharge
values along the borderline between the southern Sahara and the Sahel.

### 360    3.2.2 Spatial differences between models at $0.5°$ spatial resolution

The recharge maps generated by the RF models demonstrate a higher level of spatial detail on regional scales than the LMM-
derived recharge map (Figure 2). Recharge values predicted by RF models vary more significantly in one region (e.g. in
the tropics) whereas the LMM predictions are smoothed out by kriging interpolation. Such a high level of spatial variability
in recharge can be expected in the tropics due to the inclusion of a greater number of high-resolution explanatory factors:
precipitation, soil moisture, aridity index, potential evapotranspiration and NDVI, as the variability in recharge is associated
with the variability in the predictors, especially since there are no observations from this region that could constrain it.

The highest absolute differences in recharge estimates (Figure 5) occur south from the Equator, in the tropical and humid
wet-and-dry climate zones. The RF model predictions are more than 75 mm/yr higher than the LM model values. Other
areas exhibiting similarly high differences are the Ethiopian Highlands and the western part of Madagascar. The high relative
difference in modelled values in the Sahara is simply due to the difference between very low recharge values predicted by the
RF model (<1.5 mm/yr) and negligible LM model-derived recharge (<0.1 mm/yr). The underestimation of recharge values by
the RF models in comparison to the LM model along the borderline between the southern Sahara and the Sahel is caused by the
fact that the RF recharge values increase more gradually when moving towards the Equator. The most substantial difference in
both absolute and relative recharge is found in Angola and in the southern part of DR Congo, in the tropical savanna climate.
Here, RF model-derived values are twice as large as the LM model predictions (160-240 vs 80-120 mm/yr). However, no
observations are available from this region to assess the accuracy of these estimates.

The effects of residual kriging added to the base RF model are very localised (Fig 5c and 5g), leading to a correction of
RF-predicted values and pixels within a small radius around the fitted values (180-200 km). This indicates that the base RF
model at $0.5°$ can capture most of the spatial variation in the observational data. The difference in predicted LTA recharge

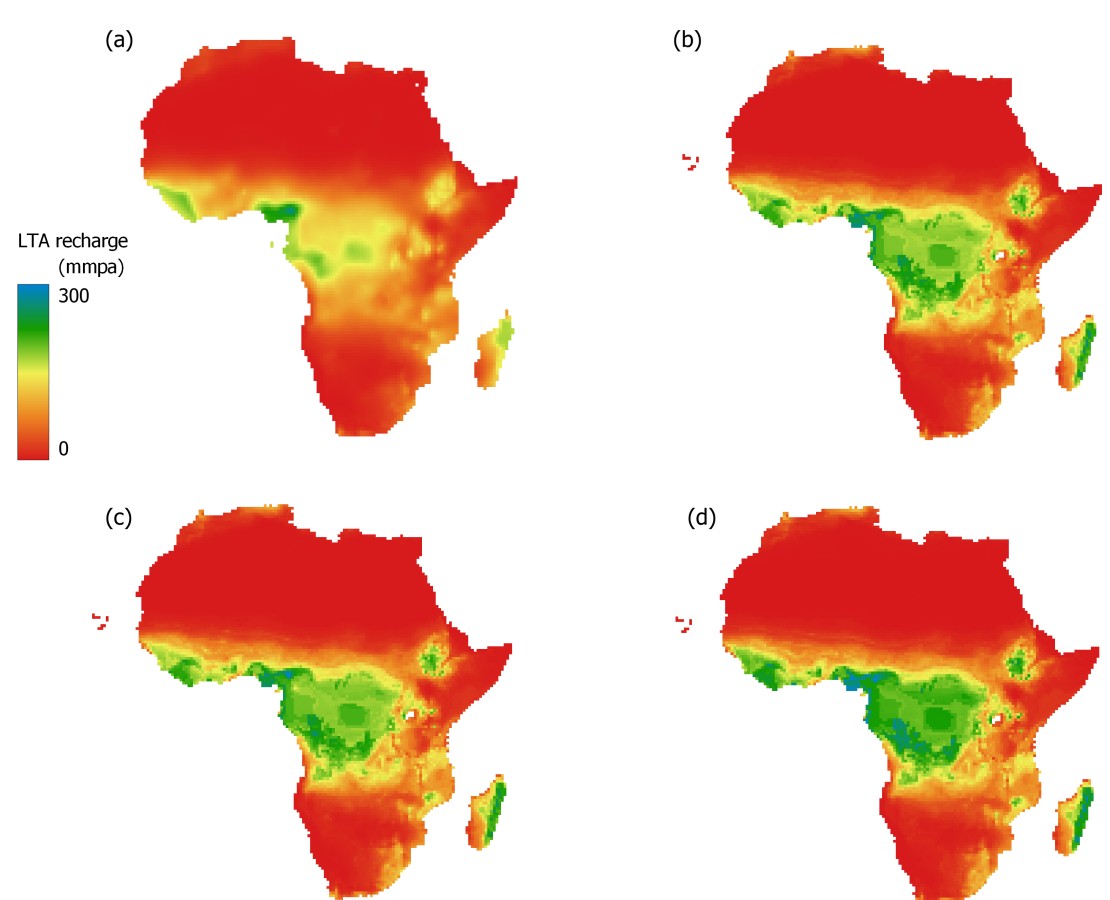

**Figure 2. Comparison of LTA groundwater recharge maps for continental Africa** at a spatial resolution of $0.5°$, obtained using a linear mixed model by MacDonald et al. (2021) (a) and three variants of random forest model: base random forest model (b), random forest with additional residual kriging (c) and random forest applied to a dataset extended by sample points from zero-recharge sampling sites (black dots) (d). Differences between these models are detailed in Section 3.2.2



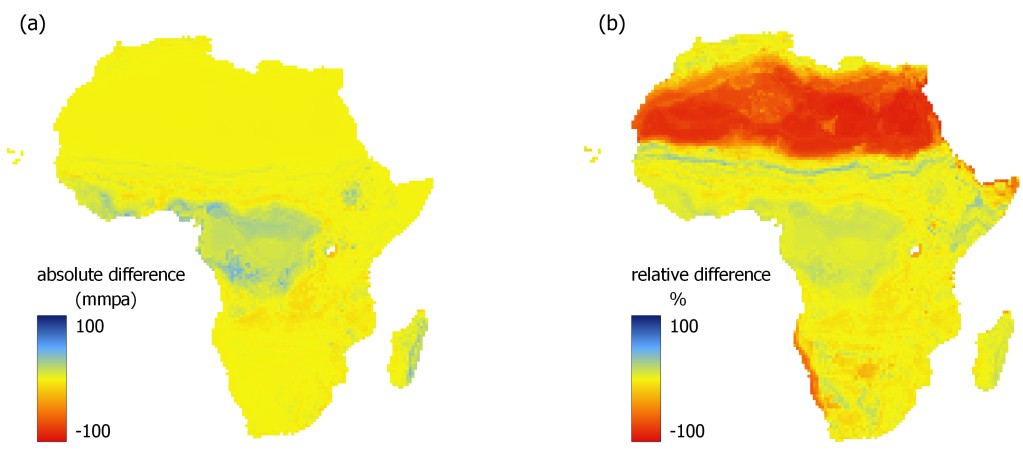

**Figure 3. Effects of including zero-recharge observations on continental LTA recharge maps generated using random forest**. Absolute (a) and relative (b) spatial differences between groundwater recharge maps for continental Africa, obtained using two variants of random forest model (without and with zero-recharge observations - Figure 2b, 2d) at the spatial resolution of 0.5°. The difference was calculated as follows: *RF (zero-recharge sites included) - RF (base)*.

ranges between ±17 and 39%, depending on the sample point (e.g. Burkina Faso: from 29 to 44 mm/yr, +39%; Libya: from 1.3 to 0.8 mm/yr, -38%; Cameroon: from 265 to 334 mm/yr, +26%).

### 3.3    Modelling groundwater recharge across Africa at 0.1° spatial resolution

### 3.3.1    Random forest-based LTA groundwater recharge maps and comparison with 0.5° model

Recharge maps at the spatial resolution of 0.1° obtained employing a RF model with and without residual kriging, built
using the explanatory factors (Table S1, Supplementary Material) and the corresponding optimal hyperparameters (Table 1) is illustrated in Figure 4. Apart from a higher resolution, these models differ from the models at the spatial resolution of 0.5° primarily by employing predictor datasets from other sources and by using recalculated hyperparameters. At 0.1°, the influence of additional residual kriging is also very localised. The fitted residual variogram demonstrates only a slightly higher spatial dependence of residuals up to around 200-250 km. The predictive performance of both versions of a RF model, in terms of
the $R^2$ value in the log scale, is comparable to the performance of the models at 0.5° (Table 3). The RF model with residual kriging explains 7% more variance than the base RF model.





**Figure 4. LTA groundwater recharge modelled at a spatial resolution of 0.1°** using (a) linear mixed model with residual kriging, (b) base random forest and (c) random forest with residual kriging.



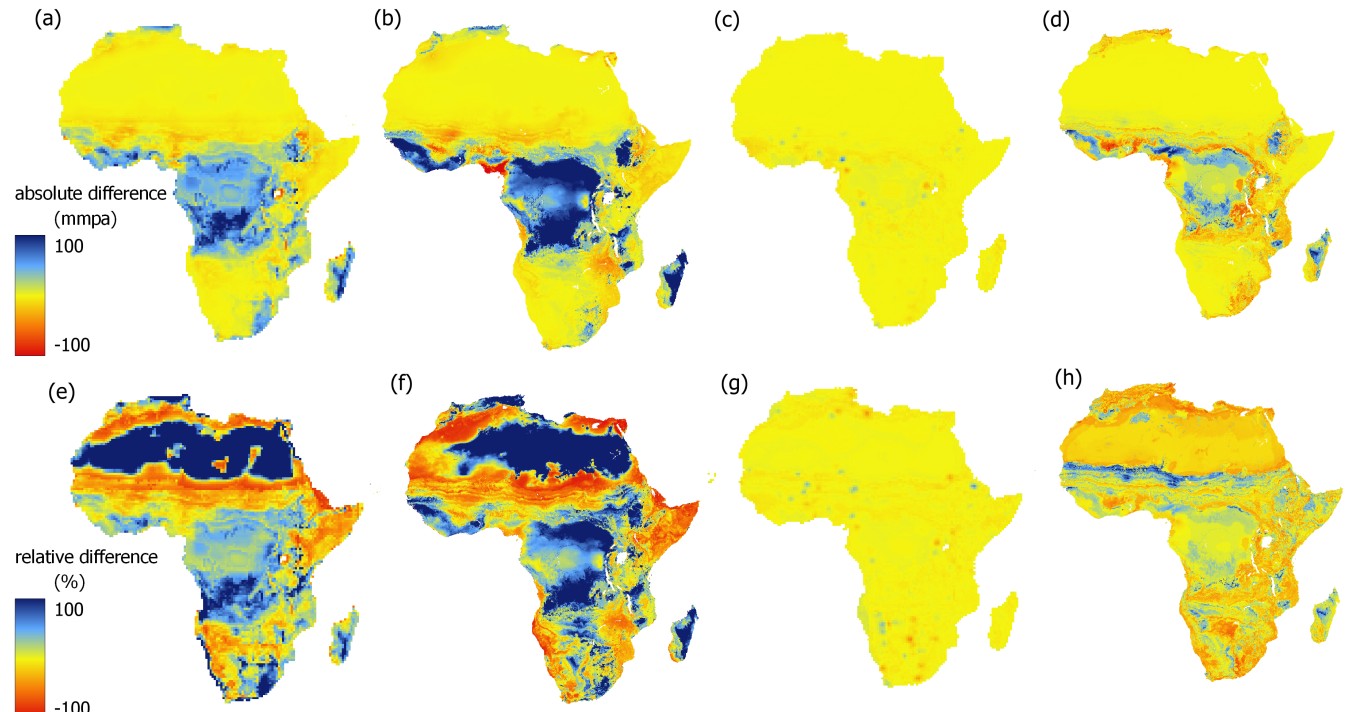

**Figure 5. Absolute (top row) and relative (bottom row) spatial differences between groundwater recharge maps for continental Africa**: (a)/(e) random forest model with residual kriging and linear mixed model at 0.5 ° *RF_RK - LMM*, (b)/(f) random forest model with residual kriging and linear mixed model at 0.1 ° *RF_RK - LMM*, (c)/(g) random forest model with residual kriging and base random forest model at 0.5 ° *RF_RK - RF*, (d)/(h) random forest model with residual kriging and base random forest model at 0.1 ° *RF_RK - RF*.

Moving from 0.5° to 0.1°, several substantial differences in regional groundwater recharge are evident using both RF models. In the most humid areas of West Africa and the Gulf of Guinea, the modelled recharge reached 250-350 mm/yr. Also, a significant increase in the recharge in the Eastern part of DR Congo was modelled (from 160-180 mm/yr to 220-260 mm/yr). Notably, high recharge values were predicted in Morocco, reaching up to 120-170 mm/yr, compared to 10-30 mm/yr modelled by the lower resolution RF models. A possible anomaly in the soil moisture dataset around the South-African cities of Pretoria and Johannesburg (0.4 m$^3$/m$^3$ vs 0.24 m$^3$/m$^3$ in the neighbouring cells) led to a high local recharge estimate. However, high soil moisture content in the highly urbanised areas might be linked to extensive irrigation and therefore cause elevated recharge rates.

### 3.3.2 Linear mixed model-based LTA groundwater recharge map and comparison with 0.5° model

The LMM at 0.1° scores a marginally better R$^2$ than the original LMM at 0.5° by MacDonald et al. (2021) (0.92 vs 0.86; Table 3). The spatial patterns are unchanged which is expected, as both LM models rely on precipitation datasets of similar spatial distributions (CRU TS at 0.5° and CHIRPS at 0.1°) - see the additional precipitation sensitivity analysis in Section 5b





of Supplementary Material. With a higher resolution precipitation dataset, more small-scale details become visible (Figures
2a and 4a). For example, LTA recharge values in Morocco are twice as high at higher resolution: 20-30 vs 60-70 mm/yr; in
the East DRC: 120 vs 240 mm/yr; in the Republic of Congo: 190 vs 250 mm/yr. A decrease in predicted LTA recharge is
found in some places like Mozambique and Madagascar, where no observations are available. Also, at the higher resolution,
exceptionally high observed recharge values lead to the creation of distinct spikes in localised, considerably higher modelled
recharge, e.g. the observations in Cameroon (obs. 941, pred. 469 mm/yr) and DRC (obs. 420, pred. 250 mm/yr).

### 3.3.3 Spatial differences between models at 0.1° spatial resolution

At 0.1°, spatial differences between all models are visibly more prominent than at the lower resolution, as shown in Figure 5.
The addition of residual kriging to the base RF model leads to higher variation in predicted recharge values in the entire domain
(Figure 5h). The biggest absolute changes are present in the humid areas of Central Africa, almost symmetrically around the
Equator. These increases in results of the extended RF model may be driven by high residual values from two observations: in
Cameroon and DRC/Republic of Congo. Other significant increases occur in Ethiopia and West Africa. Decreases are driven
by negative residuals primarily in Morocco, Ivory Coast and along the west edge of East African Rift.

Similarly to the results at 0.5°, RF models predict higher recharge rates in Northern Africa than the LMM. The extraordi-
narily high recharge observation in Cameroon drives a high local anomaly in DRC due to a high residual in the LMM, which
is more localised in the LMM than in RF models. The LMM-derived map resembles closely the input precipitation map, as
precipitation is the only explanatory factor whereas RF models also incorporate signals from the remaining employed predic-
tors: soil moisture, NDVI, aridity index and PET. There is therefore a significant difference in recharge predictions in Central
Africa, where there are no observations to constrain the models.

## 4 Discussion

### 4.1 Random forest (RF) vs Linear mixed model (LMM)

The results of this study confirm that the RF technique is able to model the LTA groundwater recharge with an accuracy com-
parable with the linear mixed model by MacDonald et al. (2021). The overall fit of both LMM and RF models to observations
was comparable, as indicated by high $R^2$ values. It confirmed that a LMM based only on precipitation was able to perform on
the observational set as well as a more sophisticated RF model driven by 5 variables. When modelling groundwater recharge on
a continental scale, the RF model resulted in a considerably higher spatial variability of the recharge, especially in the tropics.
This was expected given that very high spatial variability is also modelled in other parts of the world (Shamsudduha et al.
(2015)), and that there is a significant precipitation variability in the tropics, as confirmed by the variability in the rainfall-
correlated datasets (precipitation, soil moisture, NDVI). The ability of the RF model to capture a small-scale variability in the
input datasets and to mirror it in the recharge predictions, given sparse observations, is a clear advantage over the LMM. The





latter produced a recharge map with values that are smoothed out through kriging interpolation, consequently losing the degree
of detail of the high resolution predictor datasets.

Apart from the improved spatial level of detail, the RF model was able to detect additional areas of recharge such as coastal
regions of Morocco and Algeria. Other substantial spatial differences occurred in all climatic zones, especially south of the
Equator and in South-East Africa. This was possibly caused by the extended predictor set and by different model structure.

### 4.2   Effects of adding residual kriging

Additional residual kriging on top of RF-based recharge estimates does not have a significant effect on the prediction at
both spatial resolutions. Compared with the residual variograms of the LM models, RF-based residual variograms exhibit
considerably smaller semi-variance, which indicates that the base RF model captures the variance in LTA groundwater recharge
predictions well, based only on the predictor dataset. There is no strong smoothing effect as seen in the LMM.

### 4.3   What insight do we gain from higher resolution?

The observed differences in the low and high resolution LTA groundwater recharge maps have two likely origins. Firstly, as
both LM and RF models heavily depend on the employed precipitation datasets, some of the spatial differences might naturally
be caused by using rainfall data from different sources. Some global precipitation datasets exhibit significant discrepancies in
the long-term mean over equatorial West Africa because of their low gauge densities, as well as in interannual and decadal
variations in rainfall over the Congo Basin (Sun et al. (2018)). In this analysis, regional rainfall discrepancies could have
induced small-scale differences that became apparent in both LM and RF models, when comparing their results at different
resolutions. Both data-driven models, as opposed to process-based models, require careful input selection and quantification
of uncertainties in the input dataset, as the quality of output can only be as good as the quality of the input dataset. Secondly,
differences could also result from small-scale variability in predictor datasets at a very high resolution. Due to the ability
to retain a high spatial variability in predictor datasets (despite sparse observations), the RF model could be used for an
extended study that incorporates observations of focused recharge. Such an approach, combined with high resolution data on
the occurrence of surface water and more advanced soil and aquifer related predictors, could identify areas of small-scale,
focused recharge that largely contributes to aquifer replenishment in drylands (Cuthbert et al. (2019); Seddon et al. (2021)).

### 4.4   Bias towards dry regions

It is apparent from the LTA recharge map generated using regression combined with residual kriging that the sample of ob-
servations is not sufficient to create a model that generalises well for observations in the humid regions. High residuals are
present for the great majority of sample points in the tropical wet regions of high aridity index (e.g. observations in Uganda,
Burundi, DRC, Cameroon and southwest Nigeria), where the difference amounts to 25-60%. These residuals lead to localised
spikes in the predicted recharge around the fitted values, mostly visible in the LMM-derived maps. Due to the very limited
number of observations in the tropical humid regions, the models are not well constrained at high mean precipitation rates,





which is also reflected in the high uncertainty in these areas. The LTA groundwater recharge predictions are very likely largely
underestimated. Also, high number of observations from drylands dominates the training of the RF model and fitting of the
LMM, leading to the conclusion that climatic and meteorological factors can explain the large majority of variance. Other
topographic and geological factors, alongside a higher number of observations in tropical humid areas, need to be incorporated
in the models to better predict higher recharge rates.

### 4.5    Uncertainty and sensitivity of random forest model

Regional recharge estimates should be interpreted with caution, as they highly depend on input data quality and on the spatial
distribution of observations. Both RF and LMM models were sensitive to the changes in input datasets, especially in the areas of
limited observations. In addition, the inclusion of the Saharan zero-recharge points had a substantial impact on the groundwater
recharge predictions by the RF model in other areas that lack observational data such as: in Central Africa, the Horn of Africa

and Namibia. The effects of additional residual kriging are visible across the whole domain, including the areas where no
observations could constrain corrected predictions.
     Additional potential stability issues were observed during the supplementary variable selection process. As the RF model
is characterised by a data-dependent tree structure, the composition of the training and testing datasets had a large impact on
the model performance. In some extreme cases, the model was unable to generalise from the training samples to the unseen

(testing) data, as indicated by a disparity of the $R^2$ values in a series of RF models calculated in that process (see e.g. Table 2).
All these results highlight a possible problem with the suitability of the model in this domain with the current training dataset of
groundwater recharge observations. The observational data are highly skewed to a few high recharge samples, which led to high
model residuals at these points. This is a fundamental problem of the loss of prediction accuracy with the increasing observed
values, and the incorporation of additional high-recharge data could possibly improve the performance and the stability of

the model and constrain high differences in the predicted recharge in the observation-sparse equatorial regions and reduce the
current bias towards drylands. Nevertheless, the RF technique was able to reproduce the results of the LMM by MacDonald
et al. (2021), which indicates that further prediction improvement depends heavily on the quality of input data and the inclusion
of more observational points.
     Contrary to other ML algorithms, the RF model does not require extensive hyperparameter optimisation, so that only a

small performance gain can be achieved through model tuning (Probst et al. (2019a)). Although this was indeed the behaviour
observed when comparing the performance metrics before and after the hyperparameter search in this study (Table 2), hyper-
parameter tuning led to a substantial reduction in predicted recharge in some regions where no observations were present, as
evident from Figures 2b (with model tuning) and S8a - Supplementary Material (without model tuning). In some cases, the
difference reached up to 150 mm/yr.

A similar situation was detected when comparing the influence of various precipitation datasets. Although the spatial dif-
ferences between the CRU and CHIRPS precipitation datasets were relatively small and the other predictor datasets remained
mostly unchanged apart from their resolution, the recharge values obtained at the spatial resolutions of 0.1° (Figure 4b) and
0.5° (Figure 2b) differed significantly not only in the humid areas along the equator but also in the semi-arid regions of North-



West Africa. A sensitivity analysis to model hyperparameters would be a natural extension to this study and investigate the extent of uncertainty in the modelled recharge linked to hyperparameter tuning. Also, a more sophisticated hyperparameter search technique could be employed in the future (Probst et al. (2019b)). However, the inclusion of more groundwater recharge measurements would likely constrain the variance in the modelled values caused by hyperparameter tuning.

The RF technique makes no assumptions about the distribution of the underlying data and thus is non-parametric. Although it can deal with a limited sample size (Scornet et al. (2015)), some research applications need very large datasets to achieve accurate predictions (e.g. for medical prediction problems, van der Ploeg et al. (2014)). In the context of continental-scale long-term groundwater modelling with sparse and unevenly distributed observations, the observed variance in the modelled recharge values across the continent suggests that the current dataset size and sample distribution leads to high uncertainty. This suggests that a more diverse and larger training dataset that is better able to represent the climatic and geological diversity and the size of the study area might achieve significantly better results.

## 4.6 Future work

The new RF model could be extended by other explanatory factors and more measurements in high recharge areas in order to improve the fit to the high recharge observations that may involve locations of focused recharge; the distribution of which cannot be explained by the variability of the currently employed climatic and vegetation predictor variables. Also, further sensitivity analysis of the model to the hyperparameters and to the dataset size could be performed to better quantify the uncertainty identified in this study.

Continental modelling results at both spatial resolutions can be compared directly with the output from large scale process-based models such as WaterGAP (Müller Schmied et al. (2021)) and PCR-GLOBWB (Sutanudjaja et al. (2018)). When employing a RF model to an observational dataset from a different time window, other variables (e.g total terrestrial water storage) can be employed to test the model, such as GRACE satellite data (e.g. Bonsor et al. (2018); Scanlon et al. (2022)), which have been employed to identify inaccuracies in global hydrological model-derived decadal trends in terrestrial water storage (Scanlon et al. (2018)).

More importantly, a modified approach to input data preparation could significantly improve model accuracy. West et al. (2022), based on the work of Winter (2001), propose large-scale regionalisation and incorporation of the concept of Recharge Landscape Units (RLU) to group similar areas (in terms of climatic, landcover/use, topographic and geological features, occurrence of perennial and ephemeral water bodies). Such an approach could capture important groundwater recharge factors that dominate locally, within a specific hydrogeological setting. As the observational dataset used in this study is biased towards arid and semi-arid regions, the use of the entire dataset for the continental recharge modelling without explicitly taking into consideration distinct regional differences and intracontinental diversity (tropical and humid vs dry, upland vs lowland) might effectively ignore significant local drivers. Although West et al. (2022) concludes that grouping based on the applied global datasets of selected predictors is insufficient for explaining the variability within individual RLUs, the combination of the concept of RLUs, comparative hydrology and machine learning could improve large-scale LTA groundwater recharge estimations;



use of samples from other regions classified as similar RLUs would increase the number of observational points to train the algorithm and could potentially reduce the current bias towards drylands.

## 5   Conclusions

In this study, a random forest (RF) model was developed for the first time to predict long-term average groundwater recharge on a continental scale. It is capable of reproducing the results of a linear mixed model (LMM) developed by MacDonald et al. (2021) in terms of the model fit to the groundwater recharge dataset compiled in the aforementioned study. At a spatial resolution of $0.5°$, the key advantage of the RF model over LMM is its greater ability to capture high spatial variability in the input dataset and to mirror it in the predicted recharge values. In this way, it has been possible to identify the areas of recharge

that were previously unrepresented (e.g. in North-West Africa), and to map the variability in recharge at small scales, which is expected to be particularly characteristic of humid regions.

The use of the input dataset at a finer spatial resolution of $0.1°$ enabled the generation of a high resolution continental recharge map that could inform country-level groundwater management decisions, and support testing and calibration of mechanistic global hydrological models. However, the results should be interpreted with caution, primarily in regions of sparse

observational data, due to high model uncertainty. This limitation was identified through the use of precipitation data from different sources, tuning of hyperparameters and the inclusion of zero-recharge sample points that were initially excluded from the analysis due to the assumed lack of causality between the precipitation and the groundwater recharge. The addition of residual kriging to the base RF model slightly improved its fit to high recharge observations, though the applied predictors: precipitation, potential evapotranspiration, aridity index, NDVI and soil moisture were unable to explain those high recharge

values. High residual values for the data points located in the Gulf of Guinea and in DR Congo suggest that the model is not able to create a link between the current set of predictors and the few very high recharge observations. The incorporation of yet unidentified factors representing subsurface heterogeneity may improve the prediction accuracy of the model for these points and effectively allow to incorporate focused recharge that is crucial to assess accurately the renewability of groundwater resources in the semi-arid regions, but it would require a significant amount of data engineering. Future work should also aim

to incorporate the concept of Recharge Landscape Units, which could help to capture the variability in dominance of individual explanatory factors across different hydrogeological environments. In general, the inclusion of more LTA groundwater recharge sample points, especially from tropical humid regions, could improve model predictions and reduce the current bias towards drylands in the input dataset. As the interest in groundwater resources in Africa is growing due to their resilience to short-term climatic variability, more groundwater recharge surveys are expected to be conducted across the continent, which

will allow the full use of RF's predictive potential.

*Code and data availability.*  The output maps are available in the form of a georeferenced TIFF: Pazola (2023), *https://doi.org/10.6084/ m9.figshare.22591375.v1*. The code used for their generation (R and Python) is publicly accessible on Github: *https://github.com/pazolka/*



*rf-groundwater-recharge-africa*. Input datasets derived from previously published material: *https://doi.org/10.6084/m9.figshare.22591438.v1*.

*Author contributions.* AP conceived the idea to apply machine learning to develop continent-wide predictions of groundwater recharge. AP, MS, RGT and JF co-developed the methodological approach. AP wrote the first draft of the manuscript. AP, RGT, MS, JF, AMM, TA and IBG contributed to writing and editing the manuscript.

*Competing interests.* The authors declare that they do not have any competing interests to report.

*Disclaimer.* Copernicus Publications remains neutral with regard to jurisdictional claims in published maps.

*Acknowledgements.* We would like to acknowledge provision of datasets of model predictors and of observed long-term average groundwater recharge by the British Geological Survey. BGS recharge data and model outputs are available online: MacDonald et al. (2020), *https://doi.org/10.5285/45d2b71c-d413-44d4-8b4b-6190527912ff*. AMM, TA, IBG, and RGT acknowledge support from the UPGro research programme (Grant Numbers NE/L001926/1, NE/M008932/1, NE/M008606/1) co-funded by the Natural Environment Research Council (NERC), UK Foreign, Commonwealth & Development Office (FCDO), and the Economic and Social Research Council (ESRC). RGT is a
CIFAR Fellow in the Earth 4D: Subsurface Science and Exploration Program.



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
