# Peer review of "High-resolution long-term average groundwater recharge in Africa estimated using random forest regression and residual interpolation"

_EGUsphere, 2023_

## Referee Comment (RC2)

**Referee comment to:**

EGUsphere-2023-1898 Revision report

"High-resolution long-term average groundwater recharge in Africa estimated using random forest regression and residual interpolation" by Pazola et al. (2023).

**General Comments**

Pazola et al. (2023) provide an interesting machine learning and residual interpolation for groundwater recharge mapping at the continental scale of Africa.

The authors have used machine-learning called random forest model to estimate groundwater recharge across Africa. In addition, their models explore the potential factors affecting groundwater potential.

The paper is interesting and within the scope of the EGUsphere journal. In general, machine learning is well-placed in EGUsphere. The authors have done very diligent work by summarizing many publications applying machine learning and linear mixed models. The manuscript can be interesting to the scientific community working on machine learning applied in hydrology. The manuscript is very well written and we thank the authors for adding the codes, however at the present state; I would not recommend it for publication because certain comments need to be addressed again.

- The introduction is well. It should be worked out why this study with machine learning is necessary, knowing that machine learning is a "Blackbox model" and what its benefit is with other methods such as fuzzy logic, the frequency ratio, weight of evidence, or multi-criteria decision analysis (MCDA). The overfitting problem is one of the drawbacks that affect the accuracy of models in machine learning. Why did decide to choose the Random Forest compared to LLM models? It would be interesting if you compare machine-learning models and physics-based models to estimate groundwater recharge.

- Can you explain why the choice of the period of modelling 1981-2010? Because the input data in Table S1 has multiple Periods.

- Did you limit the validation of the random forest model with cross-validation? Alternatively, do you have the intention to integrate the external validation by compiling local raw data?
- The authors need to highlight deep the uncertainty in GIS data resampling. According to the authors what was the influence of the data resampling (0.5° spatial resolution and 0.1° spatial resolution) in the different models (LLM and RF models)?
- We know that RF is robust against the multicollinearity of features. Did you try to test the multicollinearity of predictive factors? If not, please can you use the variance inflation factor (VIF) and tolerance (TOL) indices as are customarily used to estimate the multicollinearity of all predictive factors in machine learning modelling? For example, we think that Precipitation and ET are not a problem for parameter estimation because Aridity is based on P and ET. Can you give more explanations?

- Can you explain to us the difference between the final variables in your random forest model compared to the variables selected in the study of Moeck et al. (2020)? *A global-scale dataset of direct natural groundwater recharge rates: A review of variables, processes and relationships*. https://doi.org/10.1016/j.scitotenv.2020.137042. Please cite this reference in your study.

- What is the effect of training dataset sample size on the performance/quality during the implementation of the RF model?

- Did you try to make a sensitivity analysis of the effect of each factor (explanatory variables) on the groundwater recharge map, i.e., when you decide to eliminate one or more factors?

- We know that the various GIS layers come with different spatial resolutions. Why did you choose to develop the final map at 0.1° spatial resolution? Can you explain the choice of this type of resolution?

- Do you have performed/checked quality of GeoTIFF datasets before the modelling?

- In the discussion, the authors must address the uncertainty in the GIS explanatory dataset used to estimate groundwater recharge (deficiencies of data quality; biased and absent data, sample sizes, missing covariates, etc.).

- Is it possible to improve the performance of the random forest model developed in your study? Which additional predicting variable (s) (even if such information is scarce) could be added to improve the results?

- Why you did not test the continental scale model at the country level/scale by using the best variables retained in your final model? In others words, Can you validate your machine learning model at the local scale?

- **Abstract section:**

Line 10: Put semicolon ";" between 0.83 and 0.88

**Specific comments:**

**Page 2:**
Line 23, replace ~ by the word approximatively.
Line 28. Add, "s" in the word "contributes".

**Page 3:**
Line 76. Add the article "a" in this sentence "A recent study by Huang et al. (2019) employed a multi-layer perception network"…………
Line 88. In this sentence, "In the field of groundwater modelling. The RF technique has… please check the *dot* between modelling and The RF.

**Page 4.**

Line 92. It may be interesting to show the equivalent of the spatial resolution like 0.5° in terms of distance (km) for more appreciation.

Line 107. Add the term 'the two" before "different models.

Line 108. We think that this paragraph "*Section 2 summarises the study area and the spatial characteristics of its groundwater resources, and outlines the data sources and the model development process. Section 3 presents the results of the modelling experiments. Section 4 discusses these results in the wider context and critically evaluates the developed model*" is not very important here and can be removed and keep just the sentence starting by "this study is accompanied by a Supporting Material that provides extensive information on the predictors used and additional analyses that extend the investigation presented in this paper.

**Page 5**. The study area section is not clearly presented. For example when the authors say that: "*These provide a basis for the division of the continent into 8 climatic regions, most of which experience high interannual rainfall seasonality*". We need to present clearly with a little section these 8 climatic regions. Please improve this section.

**Page 6**.

Make sure all your Figures are correctly inserted. Because, for example, the map of Figure 1 cuts the sentence in Line 151.

Line 160. Add "The" before number of wet days.

**Page 7**. Line 169. Just say: To create the groundwater recharge map…

**Page 12 and Page 13**. Add some reference to justify your finding in semi-arid and arid context results such as Burkina Faso, Ethiopia, etc. Please Line 327 to Line 356.

**Page 12**. Insert the Table 1. Optimal random forest hyperparameters found through random search with cross-validation for different random forest model variants used in this study at the end of this sentence: " *The model underestimates these samples (136 obs/38 pred, 221 obs/64 pred, 266,…*"

**Page 16**. Again, a map of Figure 3 divides the sentence in Line 380. Need to be arranged.

**Technical correction**

**Page 6**. Line 160. Please put a space between 300 and meter.

---

## Author Comment (AC1)

**RESPONSES TO REVIEWER 1:**

*The authors use an RF approach to generate spatial long-term average groundwater recharge for Africa based on 134 recharge values from the literature and compare their results with the field observations and a previous publication using an LMM (linear mixing model). The results are generated and compared for two spatial resolutions. The RF approach is very similar to LMM but offers a higher spatial variability than LMM and therefore also shows small-scale trends.*

*Even though the approach is generally ok, the manuscript is very well written and the workflow and code(s) is available through github (which I really appreciate), I still have some critical points that should be considered and discussed in detail in a revised version.*

*I'm somewhat unsure about the better spatial resolution of the results. Just because the resolution is better doesn't mean the results are more reliable. There is a very large uncertainty due to the few observations and their distribution but the maps suggest a much better and more robust result and this is dangerous. What would be the next step with the results or what can the better spatial resolution be used for? If the data is extracted directly from the maps (for water budget calculations, for example) this can lead to very distorted results, as the simulated recharge values are very uncertain for many areas. I believe the whole uncertain should be better discussed and the maps must better highlight the uncertainties (maybe with transparent colors, see my comment below)*

**R1 => We thank Reviewer 1 for their positive comments on our manuscript and note their concerns over the possible inference that high-resolution predictions are more robust. We do not claim that higher resolution data are more robust yet understand that this potentially could be implied, especially given comparison with predictions of the linear mixed model that were considerably smoothed out. We address the issue of uncertainty by constructing prediction intervals for each grid cell using Quantile Random Forest (Meinshausen, 2006; Fox et al. 2020). Although RF provides information on the conditional mean of the output variable, QRF instead provides information on the conditional distribution function of the response. By providing the prediction intervals, the reader and potential user are informed of the underlying prediction uncertainty.**

**Current global hydrological models typically operate at 0.5° spatial resolution, and large-scale prediction maps like MacDonald et al. (2021) have similarly been produced at this resolution. There is, nevertheless, an on-going trend toward hyper-resolution models (e.g. 0.1°) at continental to global scales. There is thus a need for robust approaches to the development of empirically derived datasets at higher spatial resolutions to test large-scale recharge models and support recharge mapping.**

*I wonder why, for example, seasonality in precipitation is not present in the climatic input data. In some regions, precipitation only falls in a few months and therefore the processes for recharge are significantly different for conditions when precipitation is distributed*

*throughout the year. Yes, LMM or RF show a good fit /regression, but certain parameters may compensate for the missing input. Also, of course, the relative importance does not show the importance of seasonality but only because this has not been tested in the RF (although it was in the previous work using LMM, but this is not transferable directly to the RF approach).*

**R2 => Seasonality in precipitation dominates the hydrology of all modelled areas on continental Africa whether in the equatorial humid tropics, tropical drylands or sub-tropical locations. This analysis estimates recharge at annual timescales and thus does not specifically capture seasonal variability in precipitation. We thank Reviewer 1 for bringing to our attention the fact that the number of wet days is not mentioned in the manuscript. It was originally considered as an input variable but it was not selected for the final model due to its weak influence. This point is included in the revised paper (e.g. Tables S1 and S4 in the Supplementary Material). The data source for the number of wet days is Harris et al. (2020), which was used in the LMM study by MacDonald et al. (2021). After rerunning the analysis, we confirm that the number of wet days was not included in the final models due to its weak explanatory power. It showed a strong correlation with NDVI and its inclusion in the predictor set did not improve the model fit in terms of $R^2$ for training and testing datasets respectively: (1) model with # of wet days 0.93/0.79; and (2) model without # of wet days 0.93/0.81.**

*Similar for depth to groundwater table (or call it unsaturated zone thickness) which is important for recharge processes, rate and timing. How important is this input for the RF algorithm and for the process description. I also wonder why distance to rivers is not included as an (raster)input, perhaps paired with discharge rates. This would help to better capture the important process of groundwater-surface water interaction and bank filtration, which many of the authors know better than I do.*

**R3 => The observational dataset on groundwater recharge, compiled by MacDonald et al. (2021) only includes diffuse recharge points. Focussed recharge is an important recharge regime, especially in drylands (Cuthbert et al., 2019), with strong seasonality in precipitation but is not specifically reported in the dataset. Consequently, we did not include predictors related to surface water-groundwater interactions as the objective of our analysis was to compare directly the RF model to another data-driven model (LMM) by MacDonald et al. (2021). There are other possible explanatory factors that we could have been considered besides the groundwater table depth such as soil structure and vegetation but this would render differences between the RF and LM models when our aim was to compare these modelling methods.**

*Of course there is a large uncertainty in the precipitation data sets and in the timing of recharge, but wouldn't it be possible to minimize these uncertainties and also the scaling (regression is dominated by the high recharge values) significantly by using the recharge / precipitation ratio and obtain more robust results? It would be nice if this can be discussed and tested more.*

**R4 => We welcome this suggestion from Reviewer 1 to minimize uncertainties associated with precipitation datasets using a recharge/precipitation ratio (i.e. the proportion of precipitation that is converted to recharge). However, given the established non-linear (power law) relationship between recharge and precipitation (see LMM – MacDonald et al. (2021) and RF models), we see no computational advantages to employing such an approach.**

*How does the spatially uneven distribution of the observations affect the results? Wouldn't it make more sense to show only the more robust areas and show the very uncertain ones transparently? Since not all climatic conditions have been covered, would clustering be useful to minimize the spatial discrepancy and influence?*

**R5 => We demonstrated that some data points have impact on recharge predictions in different regions (e.g. inclusion of zero-recharge points located in Sahara amplifies the predicted high-recharge values in the humid regions). Therefore, such simple uncertainty indicator could be misleading as well. We cannot exclude that the opposite can be true too, namely inclusion of more high-recharge observation might have an impact on predictions in more arid regions. We also showed that the model is biased towards dry regions, as historically these areas were of interest for groundwater studies. Data scarcity in humid regions leads to high residual in predicted vs observed values. In the revised manuscript, we use Quantile Random Forest to construct prediction intervals and based on the results and provide maps visualising the prediction uncertainty.**

*Is the correlation of the aridity index with precipitation and ET not a problem for parameter estimation and generally with all estimation methods? Aridity is based on P and ET, and I wonder what is the advantage of using all three parameters? Looking at the SI, precipitation and aridity are the most important parameters, and I wonder what the results would look like if only aridity was used. When I see table S4, I wonder why the results look almost the same for training and test, even if only P us used.*

**R6 => Correlation of precipitation, ET (evapotranspiration) and AI (Aridity Index) is not an issue for the algorithm itself but it's true that these variables might altogether represent redundant input. From the point of view of the model, any of these correlated features can be used as the predictor, with no concrete preference of one**

**over the others. We decided to keep all these variables as, when used together, the fit of the model was marginally improved.**

**Regarding the data in Table S4, please see our rationale above. Precipitation explains most of the variability in GW recharge, better than Aridity Index. We checked model performance with aridity alone and the model fit in this case wasn't as good as with precipitation as the only input. There is a small improvement in the model fit when all three variables P + PET + AI are used, compared with P alone or P + PET.**

```
Predictor set — R² train (log) — R² test (log)

Precip - 0.90 - 0.74
Aridity - 0.90 - 0.61
```

*I'm not an expert on RF, but aren't the results validated using the ROC curve and sensitivity, specificity and accuracy rather than just the regression? That would be more informative about the model results and robustness instead of using only a regression, or?*

**R7 => All these concepts are reserved for classification problems. The model performance in a classification problem is assessed through a confusion matrix from which accuracy, sensitivity, and specificity are obtained from. For regression problems, different metrics are computed such as mean square error or coefficient of determination, which can show how accurately predicted values match known values; they were used in this study.**

*Line 451: Also process based models require careful input selection and quantification of uncertainties in the input dataset.*

**R8 => We agree.**

**REFERENCES**

Berghuijs, W. R., Luijendijk, E., Moeck, C., van der Velde, Y., & Allen, S. T. (2022). Global recharge data set indicates strengthened groundwater connection to surface fluxes. *Geophysical Research Letters*, 49, e2022GL099010.

Cuthbert, M. O., Taylor, R. G., Favreau, G., Todd, M. C., Shamsudduha, M., Villholth, K. G., ... & Kukuric, N. (2019). Observed controls on resilience of groundwater to climate variability in sub-Saharan Africa. *Nature*, *572*(7768), 230-234.

Fox, E. W., Ver Hoef, J. M., & Olsen, A. R. (2020). Comparing spatial regression to random forests for large environmental data sets. *PloS one*, *15*(3), e0229509.

Harris, I., Osborn, T. J., Jones, P., & Lister, D. (2020). Version 4 of the CRU TS monthly high-resolution gridded multivariate climate dataset. *Scientific data*, *7*(1), 109.

MacDonald, A. M., Lark, R. M., Taylor, R. G., Abiye, T., Fallas, H. C., Favreau, G., ... & West, C. (2021). Mapping groundwater recharge in Africa from ground observations and implications for water security. *Environmental Research Letters*, *16*(3), 034012.

McNally, A., Arsenault, K., Kumar, S., Shukla, S., Peterson, P., Wang, S., ... & Verdin, J. P. (2017). A land data assimilation system for sub-Saharan Africa food and water security applications. *Scientific data*, *4*(1), 1-19.

Meinshausen, N., & Ridgeway, G. (2006). Quantile regression forests. *Journal of machine learning research*, *7*(6).

Moeck, C., Grech-Cumbo, N., Podgorski, J., Bretzler, A., Gurdak, J. J., Berg, M., & Schirmer, M. (2020). A global-scale dataset of direct natural groundwater recharge rates: A review of variables, processes and relationships. *Science of the total environment*, *717*, 137042.

Pham, Q. B., Tran, D. A., Ha, N. T., Islam, A. R. M. T., & Salam, R. (2022). Random forest and nature-inspired algorithms for mapping groundwater nitrate concentration in a coastal multi-layer aquifer system. *Journal of Cleaner Production*, *343*, 130900.

Podgorski, J., & Berg, M. (2020). Global threat of arsenic in groundwater. *Science*, *368*(6493), 845-850.

---

## Author Comment (AC2)

**RESPONSES TO REVIEWER 2:**

*Pazola et al. (2023) provide an interesting machine learning and residual interpolation for groundwater recharge mapping at the continental scale of Africa.*

*The authors have used machine-learning called random forest model to estimate groundwater recharge across Africa. In addition, their models explore the potential factors affecting groundwater potential.*

*The paper is interesting and within the scope of the EGUsphere journal. In general, machine learning is well-placed in EGUsphere. The authors have done very diligent work by summarizing many publications applying machine learning and linear mixed models. The manuscript can be interesting to the scientific community working on machine learning applied in hydrology. The manuscript is very well written and we thank the authors for adding the codes, however at the present state; I would not recommend it for publication because certain comments need to be addressed again.*

**R9 => We thank reviewer 2 for their positive comments above and are of the view that the responses and revisions to the manuscript now warrant publication of the paper in a revised form.**

*General Comments*

*The introduction is well. It should be worked out why this study with machine learning is necessary, knowing that machine learning is a "Blackbox model" and what its benefit is with other methods such as fuzzy logic, the frequency ratio, weight of evidence, or multi-criteria decision analysis (MCDA). The overfitting problem is one of the drawbacks that affect the accuracy of models in machine learning. Why did decide to choose the Random Forest compared to LLM models? It would be interesting if you compare machine-learning models and physics-based models to estimate groundwater recharge.*

**R10 => We worked with a relatively small dataset (134 points in total) and a set of explanatory variables that are correlated with each other. These were the most important factors that influenced the algorithm choice. There are multiple algorithms that could be applied to this problem (e.g. support vector machine, Gaussian process regression, random forest, gradient boosting decision tree, XGBoost, symbolic regression) and we chose RF as it was previously successfully applied to large-scale groundwater studies (Podgorski and Berg, 2020) and to studies with datasets of a similar size (Pham et al. 2022). RF is robust to overfitting, as the final prediction is an average of predictions from multiple decision trees, each trained with a different subset of data. In addition to RF, we applied residual kriging to explicitly account for variability in LTA recharge observations around the fitted values. We note that there are published studies applying machine learning with MCDA in hydrology, but it is unclear how it could be applicable to this regression problem. Weight of evidence could be an interesting addition to assess feature importance; this is examined in this work using other methods. Frequency ratio and fuzzy logic could introduce additional value to our analysis their inclusion is beyond the scope of the current analysis.**

**Comparing ML models with physical models would likely be a valuable exercise but it constitutes a separate study beyond the scope of the current analysis as would comparisons from a set of different algorithms (listed above).**

*Can you explain why the choice of the period of modelling 1981-2010? Because the input data in Table S1 has multiple Periods.*

**R11 => The choice of the modelling period is dictated by the original input data, namely determinations of mean annual recharge from a variety of methods covering the period of 1981 to 2010. We used secondary publicly available data and we have not produced any data on our own. CGIAR-CSI data (Aridity, PET) are only available for 1970-2000. It is not possible to get average values from that dataset for 1981-2010. CGIAR-CSI is widely used and is representing long term climatic average. We decided that it can be representative for the period our analysis. Land cover is a categorical variable. The revised manuscript uses a different dataset (Historical Land-Cover Change and Land-Use Conversions Global Dataset - NOAA dataset) with a mode value for 1981-2010. When revising the input sources, we noticed an error in Table S1. Landcover data was not used to create models that generated maps at 0.5° and 0.1° spatial resolutions. Also, the number of wet days is missing (Harris et al. 2020), as it was used in the variable selection process. Table S4 is updated accordingly.**

*Did you limit the validation of the random forest model with cross-validation? Alternatively, do you have the intention to integrate the external validation by compiling local raw data?*

**R12 => An external validation is preferable, but the analysis is rooted in limited historical data compiled from local raw data from different regions.**

*The authors need to highlight deep the uncertainty in GIS data resampling. According to the authors what was the influence of the data resampling (0.5° spatial resolution and 0.1° spatial resolution) in the different models (LLM and RF models)?*

**R13 => Most predictive factors were available at a higher resolution than 0.5° or 0.1° so each input raster was upscaled using bilinear (continuous data) or mode (categorical data) resampling methods to an appropriate resolution (0.5° or 0.1°). Bilinear interpolation is a good standard technique but loss of ultra small-scale details is inevitable. The revised manuscript contains a note of uncertainty in GIS data resampling.**

*We know that RF is robust against the multicollinearity of features. Did you try to test the multicollinearity of predictive factors? If not, please can you use the variance inflation factor (VIF) and tolerance (TOL) indices as are customarily used to estimate the multicollinearity of all predictive factors in machine learning modelling? For example, we think that Precipitation and ET are not a problem for parameter estimation because Aridity is based on P and ET. Can you give more explanations?*

**R14 => By design random forest should not be affected by correlated features. We focused on prediction, not on interpretability. A detailed analysis of feature importance was outside of the scope of this analysis but could be extended using these suggestions.**

*Can you explain to us the difference between the final variables in your random forest model compared to the variables selected in the study of Moeck et al. (2020)? A global-scale dataset of direct natural groundwater recharge rates: A review of variables, processes and*

*relationships. https://doi.org/10.1016/j.scitotenv.2020.137042. Please cite this reference in your study.*

**R15 => Moeck et al. (2020) point out that recharge estimates based solely on climatic variables can be misleading and that vegetation and soil structure have an explanatory power too. It's a reasonable assumption and this could be addressed in a separate study that looks more carefully into variable importance and focus on interpretability of machine learning models.**

**In our study, variables were selected for the model to match the data used previously in the Linear Mixed Model by MacDonald et al. (2021); most of the data are the same datasets. The MacDonald data set underwent a more thorough and transparent QA to give a curated dataset using techniques only appropriate to the African environment. Interestingly although local factors in soil and geology are important in controlling local recharge as shown by the residuals in the model, they do not improve large scale continental model - as discussed in MacDonald et al. 2021. In the follow up paper from the Moeck 2020 paper, only climatic factors are used for global modelling (Berghuijs et al. 2022, https://doi.org/10.1029/2022GL099010)**

**We add these points to the discussion section in the revised manuscript, citing Moeck et al. (2020) and Berghuijs et al. (2022).**

*What is the effect of training dataset sample size on the performance/quality during the implementation of the RF model?*

**R16 => It has a marginal effect.**

```
train/test ratio 70% to 30%
Mean R2 train (log): 0.94
Mean R2 test (log): 0.63
Mean out-of-bag error: 0.63

train/test ratio 75% to 25%
Mean R2 train (log): 0.94
Mean R2 test (log): 0.60
Mean out-of-bag error: 0.65

train/test ratio 80% to 20%
Mean R2 train (log): 0.94
Mean R2 test (log): 0.61
Mean out-of-bag error: 0.65
```

*Did you try to make a sensitivity analysis of the effect of each factor (explanatory variables) on the groundwater recharge map, i.e., when you decide to eliminate one or more factors?*

**R17 => We relied on built-in feature importances of RF algorithm and investigated changes to $R^2$ metric when gradually adding factors to identify which factors have very little or no explanatory power.**

*We know that the various GIS layers come with different spatial resolutions. Why did you choose to develop the final map at 0.1° spatial resolution? Can you explain the choice of this type of resolution?*

**R18 => It's the highest possible resolution that we could obtain raster data for the FLDAS soil moisture dataset (McNally et al. 2018) (at the time of the study). Global hydrological models typically work at 0.5° spatial resolution, and large-scale prediction maps (e.g. MacDonald et al., 2021) are produced at that resolution too. We sought to demonstrate that data-driven modelling can create prediction maps at a higher resolution, given input data of good quality. We are aware that there is a high uncertainty in the predictions and do not claim that a higher resolution is better (see R1); prediction maps at a higher resolution can, however, be obtained from a similar effort as lower resolution maps.**

*Do you have performed/checked quality of GeoTIFF datasets before the modelling?*

**R19 => Yes, we checked missing values and values range.**

*In the discussion, the authors must address the uncertainty in the GIS explanatory dataset used to estimate groundwater recharge (deficiencies of data quality; biased and absent data, sample sizes, missing covariates, etc.).*

**R20 => We recognise that considerable uncertainty will exist in gridded datasets (their representativity over 100 km$^2$ to 2500 km$^2$) used to estimate recharge. We opted for high-quality, published, and peer-reviewed datasets, and their origins are outlined in Table S1.**

*Is it possible to improve the performance of the random forest model developed in your study? Which additional predicting variable (s) (even if such information is scarce) could be added to improve the results?*

**R21 => We employed the most appropriate gridded datasets available based on the necessity of inclusion. It may be possible that better, detailed representation of vegetation and soil structure variables may improve results but this constitutes a separate study beyond the aims of this one.**

*Why you did not test the continental scale model at the country level/scale by using the best variables retained in your final model? In others words, Can you validate your machine learning model at the local scale?*

**R22 => We welcome the suggestion of conducting basin-level analyses, in an area with dense observational network but are unaware of the availability of datasets currently permitting such an analysis.**

*Minor comments*

**R23 => All following suggestions were considered and implemented, unless stated otherwise.**

**Abstract section:**

Line 10: Put semicolon ";" between 0.83 and 0.88

**Page 2:**

Line 23, replace ~ by the word approximatively.

Line 28. Add, "s" in the word "contributes".

**Page 3:**

Line 76. Add the article "a" in this sentence "A recent study by Huang et al. (2019) employed a multi-layer perception network"…………

Line 88. In this sentence, "In the field of groundwater modelling. The RF technique has… please check the *dot* between modelling and The RF. 3

**Page 4.**

Line 92. It may be interesting to show the equivalent of the spatial resolution like 0.5° in terms of distance (km) for more appreciation.

Line 107. Add the term 'the two" before "different models.

Line 108. We think that this paragraph "*Section 2 summarises the study area and the spatial characteristics of its groundwater resources, and outlines the data sources and the model development process. Section 3 presents the results of the modelling experiments. Section 4 discusses these results in the wider context and critically evaluates the developed model*" is not very important here and can be removed and keep just the sentence starting by "this study is accompanied by a Supporting Material that provides extensive information on the predictors used and additional analyses that extend the investigation presented in this paper.

**We respectfully decline this stylistic request to remove this paragraph since these sentences helpfully map out the order and logic of the manuscript.**

**Page 5**. The study area section is not clearly presented. For example when the authors say that: "*These provide a basis for the division of the continent into 8 climatic regions, most of which experience high interannual rainfall seasonality*". We need to present clearly with a little section these 8 climatic regions. Please improve this section.

**Page 6.**

Make sure all your Figures are correctly inserted. Because, for example, the map of Figure 1 cuts the sentence in Line 151.

Line 160. Add "The" before number of wet days.

**Page 7**. Line 169. Just say: To create the groundwater recharge map…

**Page 12 and Page 13**. Add some reference to justify your finding in semi-arid and arid context results such as Burkina Faso, Ethiopia, etc. Please Line 327 to Line 356.

**Page 12**. Insert the Table 1. Optimal random forest hyperparameters found through random search with cross-validation for different random forest model variants used in this study at the end of this sentence: " *The model underestimates these samples (136 obs/38 pred, 221 obs/64 pred, 266,…*"

**Page 16**. Again, a map of Figure 3 divides the sentence in Line 380. Need to be arranged.

**Technical correction**

**Page 6**. Line 160. Please put a space between 300 and meter.

**REFERENCES**

Berghuijs, W. R., Luijendijk, E., Moeck, C., van der Velde, Y., & Allen, S. T. (2022). Global recharge data set indicates strengthened groundwater connection to surface fluxes. *Geophysical Research Letters*, 49, e2022GL099010.

Cuthbert, M. O., Taylor, R. G., Favreau, G., Todd, M. C., Shamsudduha, M., Villholth, K. G., ... & Kukuric, N. (2019). Observed controls on resilience of groundwater to climate variability in sub-Saharan Africa. *Nature*, *572*(7768), 230-234.

Fox, E. W., Ver Hoef, J. M., & Olsen, A. R. (2020). Comparing spatial regression to random forests for large environmental data sets. *PloS one*, *15*(3), e0229509.

Harris, I., Osborn, T. J., Jones, P., & Lister, D. (2020). Version 4 of the CRU TS monthly high-resolution gridded multivariate climate dataset. *Scientific data*, *7*(1), 109.

MacDonald, A. M., Lark, R. M., Taylor, R. G., Abiye, T., Fallas, H. C., Favreau, G., ... & West, C. (2021). Mapping groundwater recharge in Africa from ground observations and implications for water security. *Environmental Research Letters*, *16*(3), 034012.

McNally, A., Arsenault, K., Kumar, S., Shukla, S., Peterson, P., Wang, S., ... & Verdin, J. P. (2017). A land data assimilation system for sub-Saharan Africa food and water security applications. *Scientific data*, *4*(1), 1-19.

Meinshausen, N., & Ridgeway, G. (2006). Quantile regression forests. *Journal of machine learning research*, *7*(6).

Moeck, C., Grech-Cumbo, N., Podgorski, J., Bretzler, A., Gurdak, J. J., Berg, M., & Schirmer, M. (2020). A global-scale dataset of direct natural groundwater recharge rates: A review of variables, processes and relationships. *Science of the total environment*, *717*, 137042.

Pham, Q. B., Tran, D. A., Ha, N. T., Islam, A. R. M. T., & Salam, R. (2022). Random forest and nature-inspired algorithms for mapping groundwater nitrate concentration in a coastal multi-layer aquifer system. *Journal of Cleaner Production*, *343*, 130900.

Podgorski, J., & Berg, M. (2020). Global threat of arsenic in groundwater. *Science*, *368*(6493), 845-850.